# Alpha-Lipoic Acid and Its Enantiomers Prevent Methemoglobin Formation and DNA Damage Induced by Dapsone Hydroxylamine: Molecular Mechanism and Antioxidant Action

**DOI:** 10.3390/ijms24010057

**Published:** 2022-12-21

**Authors:** Kaio Murilo Monteiro Espíndola, Everton Luiz Pompeu Varela, Rosyana de Fátima Vieira de Albuquerque, Rosiane Araújo Figueiredo, Sávio Monteiro dos Santos, Nívea Silva Malcher, Pamela Suelen da S. Seabra, Andréia do Nascimento Fonseca, Karla Marcely de Azevedo Sousa, Susan Beatriz Batista de Oliveira, Agnaldo da Silva Carneiro, Michael D. Coleman, Marta Chagas Monteiro

**Affiliations:** 1Postgraduate Program in Pharmacology and Biochemistry, Faculty of Pharmacy, Federal University of Pará/UFPA, Belém 66075-110, PA, Brazil; 2Postgraduate Program in Pharmaceutical Sciences, Faculty of Pharmacy, Federal University of Pará/UFPA, Belém 66075-110, PA, Brazil; 3Laboratory Immunology, Microbiology and In Vitro Assays (LABEIM), Faculty of Pharmacy, Federal University of Pará/UFPA, Belém 66075-110, PA, Brazil; 4Central Laboratory of the State of Pará-CLSP, Belém 66823-010, PA, Brazil; 5Postgraduate Program in Neuroscience and Cell Biology, Faculty of Pharmacy, Federal University of Pará/UFPA, Belém 66075-110, PA, Brazil; 6Postgraduate Program in Medicinal Chemistry and Molecular Modeling, Faculty of Pharmacy, Federal University of Pará/UFPA, Belém 66075-110, PA, Brazil; 7College of Health and Life Sciences, Aston University, Aston Triangle, Birmingham B4 7ET, UK

**Keywords:** dapsone, alpha lipoic acid, DNA damage

## Abstract

Dapsone (DDS) therapy can frequently lead to hematological side effects, such as methemoglobinemia and DNA damage. In this study, we aim to evaluate the protective effect of racemic alpha lipoic acid (ALA) and its enantiomers on methemoglobin induction. The pre- and post-treatment of erythrocytes with ALA, ALA isomers, or MB (methylene blue), and treatment with DDS-NOH (apsone hydroxylamine) was performed to assess the protective and inhibiting effect on methemoglobin (MetHb) formation. Methemoglobin percentage and DNA damage caused by dapsone and its metabolites were also determined by the comet assay. We also evaluated oxidative parameters such as SOD, GSH, TEAC (Trolox equivalent antioxidant capacity) and MDA (malondialdehyde). In pretreatment, ALA showed the best protector effect in 2.5 µg/mL of DDS-NOH. ALA (1000 µM) was able to inhibit the induced MetHb formation even at the highest concentrations of DDS-NOH. All ALA tested concentrations (100 and 1000 µM) were able to inhibit ROS and CAT activity, and induced increases in GSH production. ALA also showed an effect on DNA damage induced by DDS-NOH (2.5 µg/mL). Both isomers were able to inhibit MetHb formation and the S-ALA was able to elevate GSH levels by stimulating the production of this antioxidant. In post-treatment with the R-ALA, this enantiomer inhibited MetHb formation and increased GSH levels. The pretreatment with R-ALA or S-ALA prevented the increase in SOD and decrease in TEAC, while R-ALA decreased the levels of MDA; and this pretreatment with R-ALA or S-ALA showed the effect of ALA enantiomers on DNA damage. These data show that ALA can be used in future therapies in patients who use dapsone chronically, including leprosy patients.

## 1. Introduction

Dapsone (DDS) is used for the treatment and prevention of various disorders, including infectious conditions such as leprosy, malaria, and *Pneumocystis jiroveci* pneumonia and *Toxoplasma gondii* encephalitis in acquired immune deficiency syndrome (AIDS) patients. Furthermore, it also is used in non-infectious inflammatory diseases, such as dermatitis herpetiformis, IgA dermatitis, rheumatoid arthritis, acne conglobata, chronic urticaria, vasculitis, leukocyte bullous systemic lupus erythematosus, brown recluse spider bite and urticarial vasculitis syndromes [1,2,3,4,5].

However, DDS therapy frequently leads to hematological side effects, such as methemoglobinemia and hemolytic anemia, and DDS molecular structure/activity studies showed that the sulfone group and nucleophilic aniline rings play important roles in its biological properties and redox mechanisms [6,7,8,9]. In addition, its metabolite, dapsone hydroxylamine (DDS-NOH), formed by hepatic cytochromes P450, mainly CYP2C9 and CYP2C19, has been identified as the main causal agent of the adverse reactions [10]. Recently, our group reported an in vitro model, where the highest DDS-NOH concentrations induced the greatest methemoglobin formation in a dose-dependent manner, whilst also inducing DNA fragmentation in a similar pattern to H_2_O_2_ [11].

In addition, several studies have shown that antioxidant compounds such as resveratrol, ascorbic acid, curcumin and alpha lipoic acid (ALA) and its dihydrolipoic acid reduced derivative (DHLA), can prevent methemoglobin formation induced by DDS-NHOH. In this regard, ALA is a powerful antioxidant, also known as 1,2-dithiolane-3-pentanoic acid or thioctic acid. It is synthesized in small amounts in the human body by mitochondrial de novo synthesis, but can also be absorbed from foods (leafy green vegetables and red meats) and dietary supplements [12]. Several studies showed that ALA acts as a potential therapeutic agent for some chronic diseases associated with high levels of oxidative stress, such as diabetes mellitus (DM) and its complications, hypertension, Alzheimer’s disease, Down syndrome, cognitive dysfunction and some types of cancer [13,14,15,16]. Structurally, ALA is an organosulfur compound derived from octanoic acid. The carbon atom at C6 is chiral and the molecule exists as two enantiomers: (R)-(+)-lipoic acid (R-ALA) and (S)-(--lipoic acid (S-ALA), and as a racemic mixture (R/S)-lipoic acid (R/S-LA) [17,18]. ALA is synthesized endogenously as the R-enantiomer, found in vegetable and animal tissues in the form of lipoyllysine (attachment of ALA to specific lysine residues). However, only R-ALA is conjugated to conserved lysine residues through an amide linkage, thus it acts as a cofactor in biological systems [18,19,20,21]. However, commercial ALA, used in dietary supplement materials and compounding pharmacies, is usually a racemic mixture of the R- and S-forms (R/S-ALA) [18]. 

ALA may be effective in both the prevention and treatment of hematological pathologies, including methemoglobinemia, due to its antioxidant potential benefits as reported by several studies [22,23,24,25]. However, its antioxidant mechanisms are still poorly understood, although they may be associated with the glutathione (GSH) system and other enzyme systems (i.e., thioredoxin reductase -TrxR and lactate dehydrogenase -LDH), which can reduce lipoic acid to DHLA in the mitochondria of mammalian cells [17]. Both the oxidized and reduced forms of ALA are potent antioxidants, but the redox mechanisms still need to be investigated [17]. Thus, in the present study, we aimed to evaluate the protective effect of racemic ALA and its enantiomers on methemoglobin induction, oxidative stress parameters and DNA damage induced by DDS-NOH using an in vitro model employing human erythrocytes and lymphocytes.

## 2. Results

### 2.1. Effect of Racemic Alpha-Lipoic Acid (ALA) on Methemoglobin Formation Induced by DDS-NOH

To evaluate the effect of ALA on methemoglobin formation induced by DDS-NOH (2.5, 5.0 and 7.5 µg/mL), erythrocytes were pre-treated with different concentrations of ALA (10, 100 and 1000 µM) for 60 min and, after these cells were incubated with DDS-NOH (2.5, 5.0 and 7.5 µg/mL) for 60 min.

Overall, as Figure 1A shows, the pretreatment with ALA was able to inhibit the methemoglobin formation induced by all concentrations of DDS-NOH. This protective effect of ALA was best seen in a concentration of 2.5 µg/mL of DDS-NOH, in which all concentrations of ALA protected the methemoglobin formation in a similar manner. However, the 1000 µM concentration of ALA was able to inhibit the methemoglobin formation induced even at the highest concentrations of DDS-NOH. Therefore, 100 and 1000 µM of ALA were adopted for subsequent experiments.

### 2.2. Comparative Effect of Pre and Post Treatment with ALA or Methylene Blue (MB) on Methemoglobin Formation Induced by DDS-NHOH

As previously shown by our group, methylene blue (MB), an antidote for anti-methemoglobinemia treatment, fully protected and reversed the methemoglobin formation induced by DDS-NOH [11]. Therefore, we compared the protective effect of the pre- and post-treatment with ALA (100µM) and MB (40 nM) on methemoglobin formation induced by DDS-NOH (2.5 µg/mL). Our data showed that ALA efficiently inhibited the methemoglobin formation in common with MB; but was not able to reverse this effect (Figure 1B and C, respectively). These data showed that MB was significantly more effective than ALA even at a concentration of MB at least 2000-fold lower than that of ALA.

### 2.3. Effect of Pretreatment with ALA on Reactive Oxygen Species (ROS) Generation Induced by DDS-NOH

We previously showed that erythrocytes exposed to DDS-NOH (2.5 µg/mL) or T-BHP (Tert-butyl hydroperoxide) had higher levels of intracellular ROS [11]. Thereby, we evaluated the effect of the pretreatment with ALA in the DDS-NOH model. Our data showed that both tested concentrations (100 and 1000 µM) were able to inhibit ROS production induced by 2.5 µg/mL of DDS-NOH (Figure 2).

### 2.4. Effect of Pretreatment with ALA on CAT and SOD Activities and GSH Production on In Vitro DDS-NOH Model

Previously, we also showed that DDS-NOH inhibited CAT activity, but not SOD activity [9]. All concentrations of ALA prevented the inhibition of CAT activity induced by DDS-NOH in erythrocytes (Figure 3A) but did not change the SOD activity (Figure 3B). In addition, the DDS-NOH did not alter GSH levels compared to negative control, but all concentrations of ALA induced an increase in GSH production (Figure 3C). 

### 2.5. Effect of ALA on DNA Damage Induced by DDS-NOH

Figure 4A–D show the effect of ALA (100 and 1000 µM) on DNA damage induced by DDS-NOH (2.5 µg/mL) in peripheral blood lymphocytes. Data showed that ALA was able to significantly attenuate DNA damage (% DNA in comet tail, tail length (%), TM and OM) induced by DDS-NOH, in common with our study with resveratrol [11]. 

### 2.6. Effect of the Pre- or Post-Treatment with Enantiomers (R-ALA or S-ALA) on Methemoglobin Formation and GSH Levels Induced by DDS-NOH

Pretreatment with all concentrations of R-ALA or S-ALA enantiomers (0.1, 1.0, 5.0 and 10 µM) was able to inhibit the methemoglobin formation induced by DDS-NOH (Figure 5A). However, only the pretreatment with S-ALA enantiomer was able to change GSH levels, whereas the higher concentrations stimulated the production of this antioxidant (Figure 5C). At the same time, in relation to post-treatment, only the post-treatment with R-ALA enantiomer inhibited the methemoglobin formation induced by DDS-NOH, increasing the GSH levels (Figure 5B and Figure 5D, respectively). 

### 2.7. Effect of Pretreatment with R-ALA and S-ALA on Oxidative Parameters Induced by DDS-NOH

The pretreatment with R-ALA or S-ALA in all concentrations of (0.1, 1.0, 5.0 and 10 µM) prevented the increase in SOD and decrease in TEAC induced by DDS-NOH; however, only 0.1 µM of R-ALA, showed a significant increase in TEAC compared to the DDS-NOH group (Figure 6A,B). Regarding lipid peroxidation, the concentrations of 0.1 and 1µM of R-ALA decreased the levels of MDA induced by DDS-NOH (Figure 6C).

### 2.8. Effect of Pretreatment with R-ALA and S-ALA on DNA Damage Induced by DDS-NOH

The pretreatment with R-ALA or S-ALA in all concentrations of (0.1, 1.0, 5.0 and 10 µM) showed the effect of ALA enantiomers on DNA damage induced by DDS-NOH (2.5 µg/mL) in peripheral blood lymphocytes. Data showed that ALA enantiomers were able to significantly attenuate DNA damage (tail length (%), TM and OM) induced by DDS-NOH (Figure 7A–D). In regard to the DNA in the tail length, only concentrations of 1 and 10µM showed a decrease in the percentage of this parameter.

### 2.9. Theoretical Antioxidant Mechanism of R-ALA and S-ALA Isomers

The values of the homo, lumo and GAP boundary orbitals are important for the interpretation of the antioxidant activity. HOMO orbitals have a greater ability to carry out nucleophilic attacks on higher energy orbitals. The GAP is related to the electronic transition, and, therefore, smaller indices, (GAP_HOMO-LUMO_), are related to the increase in chemical reactivity, and consequently the donation of electrons. The results for the antioxidant capacity show that the S isomer has higher values of HOMO (−5.56 eV) and lower GAP (4.55 eV), indicating its higher antioxidant character in relation to the R isomer (Table 1).

The electron densities in the boundary orbitals are a way of characterizing the possible interactions between electron donors and acceptors. The analysis of the HOMO orbital for the two configurations R and S shows that the regions of higher electron density are located predominantly on the disulfide ring (Figure 8).

The molecular docking method is used to interpret the molecular aspects of ligand-protein interactions during the discovery of new drugs. Molecular docking results, using a blinded approach, reveal two possible coupling regions on the hemoglobin molecule. The first refers to an adjustment region with a binding energy value of −5 kcal.mol^−1^ for both configurations. In this case, the R-ALA ligand interacts with residues asp94 and tyr24 of chain A and residues phe41 and arg40 of chain D, while the ligand S-ALA interacts with residues tyr42 of chain A and residues phe41 and his97 of chain D (Figure 9). It is possible that in this region the racemic mixture acts significantly to inactivate the receptor.

The second possible binding region is close to the Heme group, for which the binding energies of the R-ALA and S-ALA isomers are divergent, −5.4 Kcal.mol^−1^ and −4.3 Kcal.mol^−1^ (Figure 10a,b), respectively, indicating R-ALA as the major inactivator of hemoglobin. Regarding the interactions and amino acid residues involved in the formation of the ligand-receptor complex, there is a simultaneous interaction with the same amino acid residues: his-45, his-58 and phe-46, however, R-ALA interacts with the HEME group, while S-ALA interacts with the pro-44 amino acid residue. Despite being a hydrogen bond, this interaction is not enough to decrease the energy of the system. At the same time, the change in the spatial configuration of the R isomer promoted by the displacement of the disulfide ring to regions closer to the HEME group can promote a charge change in addition to facilitating pi-alkyl with the HEME group. Thus, it can be responsible for the decrease in the energy of the ligand-receptor complex and greater efficiency in the enzymatic inhibition process.

## 3. Discussion

In this study, DDS-NOH-induced methemoglobin formation as well as DNA damage evidenced the oxidative potential related to treatment with this drug. At the same time, pretreatment with ALA as a racemic mixture was able to reduce methemoglobin (MetHb) levels, as well as increase glutathione concentrations in erythrocytes and protect against DNA damage in lymphocytes. Furthermore, both enantiomers (R/S-ALA) prevented at least 50% of MetHb formation and prevented DNA damage induced by DDS-NOH, by mechanisms dependent on the oxidative balance in these cells. However, only the R-ALA enantiomer was able to partially reverse the metabolite-induced MetHb formation with increased GSH levels after post-treatment. Thus, our data show that the pretreatment with antioxidant ALA and its enantiomers can prevent oxidative stress and damage to macromolecules such as DNA, lipids and proteins induced by the DDS-NOH metabolite, by mechanisms that reset the tissue oxidant/antioxidant balance. However, the racemic mixture ALA was more effective in protecting against methemoglobin formation and DNA damage, while the R-ALA isomer was effective in protecting or partially reversing methemoglobin formation, as well as preventing DNA damage in vitro caused by exposure to the DDS-NOH metabolite. The superiority of R-ALA with respect to the S-ALA isomer may be associated with its uniform distribution, which shows a curvature defined as a “geometric” or “steric” factor that led to greater stability of the molecule and success in inhibiting or altering the oxidation process of hemoglobin, as shown in the molecular modeling.

During drug administration and metabolism, erythrocytes are one of the main cells to suffer damage caused by oxidative stress, which leads to redox reactions mainly in hemoglobin [26]. Due to limited repair mechanisms, exposure to high O_2_ concentration, and the presence of a strong oxidative catalyst and hemoglobin (Hb), the antioxidant systems found in erythrocytes may not provide adequate protection against excessive exposure to reactive species. Thus, these processes can predispose the cell to oxidative damage, leading to an increase in methemoglobin (MetHb) formation and hemolysis [27]. These hematological alterations are frequently found in patients who use DDS, including leprosy patients, even at therapeutic doses [28,29]. DDS is administered daily in doses of 50–100 mg [30], leading to serum concentrations of 0.4–1.2 mg/L of its hydroxylated metabolites, including DDS-NOH, as reported in previous work by our group [11].

In this regard, our previous studies and those of other authors using an in vitro model of erythrocytes had already shown that 2.5 μg/mL of DDS-NOH induced above 15% of MetHb [11,31,32]. MetHb is a marker of the oxidation of Hb, being formed when Fe^2+^ from Hb loses an electron forming Fe^3+^, an electron that can be transferred to an O_2_ molecule, resulting in the formation of O_2_●- [33]. This is potentiated by the DDS-NOH that binds to the ferric/superoxide anion (Fe^3+/^O_2_●-) complex of O_2_Hb, favoring the formation of new molecules of MetHb, ROS and DDS-NO. This redox cycle is continued when the ROS formed, mainly O_2_●- and H_2_O_2_, during the oxidation process can oxidize other Hb molecules leading to more MetHb [34]. Furthermore, DDS-NO can react with glutathione molecules to convert to DDS-NOH, which leads to the oxidation of other Hb molecules, a cycle that only ends when GSH is completely consumed [1,35,36].

Methemoglobin is a direct effect of pro-oxidant agents that are able to access the heme portion of hemoglobin, reacting with the iron contained in this protein [37]. In this sense, it is well established that DDS-NOH acts as a reactive molecule, capable of inducing oxidative damage, mainly in blood [7], but all in tissues and organs [38]. Our group demonstrated in vivo that the percentage of MetHb is considerably higher in individuals who are treated with DDS. In this study, the authors also observed alterations in the antioxidant status of these individuals, by measuring the activity of SOD, CAT, as well as in the levels of the main endogenous antioxidant, GSH [29]. Methemoglobin formation is primarily limited by the ability of the erythrocyte to maintain intracellular levels of GSH. Thus, hemoglobin oxidation occurs cyclically, related to hydroxylated metabolites, mainly dapsone hydroxylamine (DDS-NOH), generated during the metabolic process by N-hydroxylation of dapsone [22,27].

Thus, protection and/or reversal of methemoglobin formation and restoration of the oxidant/antioxidant balance are beneficial effects of the use of antioxidants by patients using dapsone. Previously, our research group, working with resveratrol in vitro, showed that this antioxidant protects against the formation of methemoglobin induced by DDS-NOH. However, it was effective only in the pretreatment [11], in common with the data obtained in this study with the use of ALA in the form of a racemic mixture, which contains the R isomer, which is of natural origin, and the S isomer, which is synthetic [39]. In this regard, our data also showed that R-ALA was more effective to protect and reverse methemoglobin formation and increase GSH in the system.

ALA is synthesized endogenously by the enzyme lipoic acid synthetase from octanoic acid [40] and is covalently linked to specific proteins, via lysine residues. In the Krebs cycle, ALA plays important roles in several chemical reactions, functioning as a cofactor for some enzymatic complexes associated with energy generation in the cell. In addition, in other processes it can act by chelating metals, eliminating reactive oxygen species, recycling/inducing the expression of other antioxidants, has anti-inflammatory action and can lead to activation of cell signaling, among other actions [41]. Thus, in recent years, there has been a growing medical and scientific interest in the potential therapeutic use of ALA [42].

These biological activities are linked to its structure, as ALA has two thiol groups (-SH), one on each of the C6 and C8 carbon atoms, which are connected by a disulfide bridge, which can be oxidized or reduced and also forms covalent bonds with proteins. The oxidized form, ALA, can be rapidly reduced in the body resulting in the formation of dihydrolipoic acid (DHLA) [21,43]. Furthermore, ALA contains a single central chiral carbon, C6, which is asymmetric and results in two optical enantiomers or stereoisomers: R-(+)-lipoic acid (R-ALA) and S-(- )-lipoic acid (S-ALA) [41,44]. Additionally, ALA is found in the form of supplements composed of the racemic mixture of its R-ALA and S-ALA isomers in doses of 200 to 1200 mg/day [45], or of the isolated R-ALA isomer, with doses of 200 at 300 mg [46]. The R-ALA isomer is unstable at temperatures above 49 °C, while the racemic mixture remains stable at temperatures of up to between 60 and 62 °C. In pharmacokinetic studies in healthy subjects, the R-ALA isomer presents a higher level of absorption, while the S-LA isomer helps in this percentage, preventing the polymerization of the R-ALA form [47] (SEIFAR et al. 2017). In this regard, the use of supplements with the racemic mixture is more viable in relation to the R-ALA isomer [42].

R-ALA is the naturally occurring enantiomer, synthesized in the body by the mitochondrial enzyme dihydrolipoamide dehydrogenase from ALA, while S-ALA is a by-product of chemical synthesis [12]. In addition, GSH can also catalyze the reduction of ALA, more slowly, preferentially forming S-ALA, with consumption of NADPH [47]. R-ALA, like ALA, is an essential cofactor of several mitochondrial complex enzymes that catalyze reactions related to energy production and the catabolism of α-keto acids and amino acids [19,20]. This compound covalently binds to the amino group of a lysine residue, through an amide bond and, therefore, presents itself as lipoamide [21]. Therefore, R-ALA functions as a cofactor for pyruvate dehydrogenase and alpha-keto-glutarate dehydrogenase, as well as the branched-chain alpha ketoacid dehydrogenase complex and therefore plays a critical role in glucose metabolism [48]. These qualities make R-ALA a central player in the antioxidant network, which is why ALA is used as a treatment for age-related diseases, such as diabetes and neurodegenerative diseases [49,50]. Currently, ALA is widely used as an anti-aging compound in cosmetics and as a food supplement. Although it is possible to separate the ALA enantiomers, the bioactive R-ALA and the S (-)—alpha-lipoic acid (S-ALA), the commercially available ALA is the racemate. This is because although the pure enantiomer R-ALA is the bioactive form, it is unstable when exposed to low pH, light, or heat [49].

At the same time, S-ALA has a lower binding affinity to mitochondrial enzymes, and it has been demonstrated that the mitochondrial function of rats increased with R-ALA supplementation, while S-ALA, being unable to bind to mitochondrial enzymes, caused a reduction in ATP production [51]. In addition, R-ALA also increases or maintains levels of antioxidants such as GSH and ascorbic acid [52]. In an in vitro study, R-ALA was more effective at chelating metals in human erythrocytes and preventing ROS formation than ALA and S-ALA [53]. In an animal model, R-ALA was able to improve glucose transport, absorption and metabolism in insulin-deficient rats, being more effective than S-ALA [21,54,55]. In addition, Pick [56] demonstrated that the S-ALA isomer acted as a substrate for glutathione reductase, while the R-ALA isomer showed selectivity for lipoamide dehydrogenase. As these enzymes occur in different tissues and cell compartments, this reflectsthe diversity of different effects that can be exerted by the R and S isomers of ALA. In this sense, the differences between the GSH levels obtained by the two isomers of ALA in our work can be justified by this difference in molecular targets, both in dissimilar tissues and in the same cell.

ALA is a cofactor for mitochondrial α-keto-dehydrogenase complexes and participates in S-O transfer reactions that are naturally occurring cofactors that also possesslipid regulatory properties. ALA is reduced to DHLA in several tissues [15,48] and both ALA and DHLA have acted as antioxidants in relation to hydroxyl radicals and inhibit the oxidation of lipids and proteins [57]. Cell reduction of ALA to DHLA is performed by NAD (P) H-induced enzymes, thioredoxin reductase, lipoamide dehydrogenase and glutathione reductase. Red blood cells pick up and reduce ALA by glucose metabolism; subsequently, the DHLA is released into the extracellular space, thus reflecting the activity of the disulfide reductase. Red blood cells reduced the S-isomer of ALA about 40–50% more efficiently than the R-isomer when both were present at low concentrations [58].

In a study carried out by Yamada [59], ALA action was observed both in the reduction of ROS and in the improvement in the response to oxidative stress [59]. The increase in ROS induced by DDS-NOH has been noted by our [11,29]. In previous studies, from a similar model of oxidative stress induction by DDS-NOH, ALA was able to decrease ROS levels and alter antioxidant factors. This neutralizing ability of the racemic mixture of ALA on ROS (singlet oxygen and hydroxyl radical, among others) is due to the reduction potential of the ALA/DHLA system [60,61]. In addition, ALA and DHLA act as antioxidants to directly scavenge ROS and RNOS, the chelating transition and heavy metal ions and mediate the recycling of other endogenous antioxidants as well as glutathione. ALA also modulates several signaling cascades by receptor-mediated and non-receptor-mediated processes [58].

GSH is a tripeptide synthesized from the amino acid cysteine-containing sulfur that acts as the main antioxidant in several oxidative stress response processes [62]. The main property of ALA is its interaction with GSH and its ability to recycle endogenous GSH. The activity of ALA, specifically of its reduced form, DHLA, on GSH is a result of its interaction with glutathione reductase (GSR), which contributes to the maintenance of intracellular levels of GSH.

In addition to generating GSH by reducing GSSG, ALA may also be involved in the synthesis of glutathione. The oxidation-reduction process from ALA to DHLA is an uninterrupted source of cysteine, which is the amino acid limiting glutathione production rate [63]. In elderly rats, in which GSH levels were reduced, intravenous administration of ALA (40 mg/kg body weight) restored tissue levels of GSH in the heart and brain [64]. Likewise, ALA also increases GSH levels in T cells, human erythrocytes, glial cells and lymphocytes through the induction of cysteine uptake, thus increasing glutathione synthesis both in vitro and in vivo [14,65,66]. Therefore, the ability of ALA to regenerate GSH is an important mechanism, in addition to being able to eliminate free radicals and chelate metal ions, that may better clarify its therapeutic efficacy in several pathologies [63]. ALA supplementation maintains and actually reverses age-related decline in hepatocellular and myocardial ascorbate and GSH levels, even when cells were incubated with tert-butyl hydroperoxide, a model of alkyl peroxide [17]. In addition, ALA-inhibiting enzymatic inactivation induced by free radicals could be another possible mechanism for the increase in the activities of the enzymes after the treatment [17]. These results demonstrate that ALA is an effective agent for restoring both age-related declines in the thiol redox ratio and increasing GSH levels that otherwise decrease with age. However, ALA also proved to be an effective regulator of signaling pathways, such as PI3K/Akt, p38 MAPK, ERK1/2 and JNK, all important cell signaling pathways. Another possibility as an ALA pathway would be as a pro-oxidant, with both independent and Nrf2-dependent transcriptional concentration [50,59].

In this study, the evaluation of DNA damage was performed through the comet assay, a simple and sensitive technique to detect oxidative DNA damage [11]. The rate of DNA damage is directly related to the metabolic rate and the lifetime of the organisms [67]. A variety of oxidants induce DNA strand breakage, which also inactivates a nuclear enzyme, poly-ADP-ribose polymerase, involved in the repair of DNA lesions [17]. In this sense, reactive molecules forming reactive products with pro-oxidants, such as DDS and its metabolites, can cause disturbances in genetic material. DDS-NOH has been associated with DNA damage, possibly by increasing ROS, such as superoxide anion, hydrogen peroxide, singlet oxygen and radial hydroxyl, generated directly or indirectly [11,67]. Our studies with this drug have shown elevated lipid damage in the blood of individuals undergoing treatment [29], probably due to the increase in hydroxyl radicals. Animal models also demonstrate the damage caused by DDS-NOH in both lipids and macromolecules as proteins [68] and DNA [69].

Treatment with ALA exerting a decrease in the content of 8-oxo-dG was observed in elderly rats [17]. This may be due to the action of ALA to reverse the inactivation of the poly-ADP ribosylation, thus preventing the accumulation of DNA damage [70]. In addition, ALA provides protection to plasmid DNA against oxygen-induced damage through its high free radical scavenging action [17]. The evaluation of the inhibition of molecules that act as pro-oxidants and free DNA breakage was also performed with ALA. The authors concluded that ALA can inhibit this action and thus prevent disordered DNA cleavage [70]. Thus, ALA, both in its racemic form and in isolation with its R and S forms, appears to protect against DNA damage caused by drugs such as dapsone.

There are many benefits attributed to ALA, both as an antioxidant and as a modulator of inflammation and regulator of glucose absorption; pro-oxidant characteristics of this natural compound have even been reported [59]. As most commercially available ALA supplements are a mixture of both the R and S enantiomers, questions have arisen as to the preferential absorption of one isomer versus the other. In one study, volunteers received 600 mg of R-ALA and S-ALA, and the plasma concentrations of R-ALA were 40–50% higher than S-ALA [41,71], the absorption of the latter apparently being faster than that of R-ALA. Uchida and colleagues tested the selectivity of cell membranes to ALA, R and S enantiomers, in Caco-2 and MDCK II cells and concluded that enantioselectivity is due to ALA metabolism, interfering in the differentiated pharmacokinetics between the R and S enantiomers. However, the absorption through membranes does not differ between the two forms of ALA, R and S [42]. It has not yet been established whether R-ALA, its salt form, or a racemic mixture, would be better for use in future clinical studies [18].

Finally, although the HOMO and LUMO data indicate that the S-ALA isomer shows greater antioxidant capacity than the R-ALA isomer, both prevented the formation of methemoglobin and MDA. The comparison by molecular docking, evaluating a region close to the Heme group, showed that the binding energies of the R-ALA (−5.4 Kcal.mol^−1^) and S-ALA (−4.3 Kcal.mol^−1^) isomers are distinct and that the R-ALA isomer is the main inactivator of hemoglobin. In this sense, the analysis of the interactions between the amino acid residues involved in the formation of the ligand-receptor complex showed a difference in curvature between the two isomers, which allows some similar interactions with the same amino acid residues, such as his-45, his-58 and phe-46. However, the R-ALA isomer interacts with the HEME group, while S-ALA interacts with the pro-44 amino acid residue. These findings agree with our conclusion that the R-ALA isomer has a more stable format than the S-ALA isomer in binding to hemoglobin, data that are consistent with experimental in vitro observation, which also shows a better effect in reversing the formation of methemoglobin, preventing DNA damage and stimulating DDS-NOH-induced increase in GSH.

## 4. Materials and Methods

### 4.1. Chemicals

Racemic Lipoic Acid (ALA), R-Lipoic Acid (R-ALA) and S-Lipoic Acid (S-ALA),2′,7′-Dichlorodihydrofluorescein diacetate (DCFH-DA), tert-butyl hydroperoxide (t-BHP), methanol, ethanol, dimethyl sulfoxide, Triton X-100, sodium hydroxide, sodium chloride, ethylenediamine tetraacetic acid (EDTA) agarose for routine, agarose low electroendosmosis (EEO), hydrogen peroxide (H_2_O_2_), hypoxanthine, ethidium bromide, methylene blue (MET), xanthine oxidase and cytochrome C were purchased from Sigma Chemical Com. (St. Louis, MO, USA). Dapsone hydroxylamine (DDS-NOH) was purchased from Santa Cruz Biotechnology (Santa Cruz, CA, USA). Phytohemagglutinin M. was purchased from Life Technologies (Carlsbad, CA, USA).

### 4.2. Preparation of ALA, ALA Enantiomers and DDS-NOH Solutions

ALA was dissolved in 100% ethanol as a stock of 0.2 mol/L and stored at −20 °C and diluted to the required concentrations (10, 100 and 1000 μM) with PBS 0.5 M, pH 7.4, before use. The final concentration of ethanol in the PBS was less than 0.001%. R-Lipoic Acid (R-ALA) and S-Lipoic Acid (S-ALA) were dissolved in dimethyl sulfoxide (DMSO; (CH_3_)_2_SO) before the addition of aqueous solvent to prepare working concentrations [0.1, 1, 5 and 10μm]. DDS-NOH was dissolved in methanol and stored at −20 °C.

### 4.3. Ethics Statement

This study was approved by The Ethical Committee of the Federal University of Pará, Brazil (protocol 165/11 CEP-ICS/UFPA) and informed consent was obtained from all subjects prior to sample collection and experiment commencement. Thus, all participants were informed about the aims and methods of the study, and they signed the informed consent before the start of the experiment and sample collection. Venous blood from different healthy volunteers (both sexes and ages of 20–45 years) who abstained from alcohol and tobacco was obtained by venipuncture in heparin (5000 IU/mL) after an overnight fast.

### 4.4. Preparation of Erythrocyte Suspensions

The blood was centrifuged at 3000 rpm for 10 min at 4 °C. After removal of plasma, the buffy coat was removed and the isolated erythrocytes were washed three times with cold phosphate-buffered saline (PBS; 0.9% NaCl, 10 mM Na2HPO4, pH 7.4). The packed red blood cells (RBC) obtained were suspended (at 40% hematocrit) in the same solution. 

### 4.5. Pretreatment of Erythrocytes with ALA, ALA Enantiomers, or MB and Treatment with DDS-NOH

The protective effect of ALA and its enantiomers in the methemoglobin formation and oxidative stress generation was evaluated by pre-incubating of erythrocytes suspension with ALA (10, 100 or 1000 μM) or R/S-ALA (0.1, 1, 5 and 10μM) for 60 min at 37 °C as described by McMillan et al. [72]. Erythrocytes were then exposed to DDS-NOH (2.5, 5.0, or 7.5 μg/mL) for an additional 60 min at 37 °C, as per Pandey et al. [73]. In addition, to assess the effect of MB, the erythrocytes suspension was pre-incubated for 30 min with this substance (40 nM) and it was exposed to DDS-NOH, as per Reilly et al. [11,74]. Cellular viability was analyzed prior to and after incubation.

### 4.6. Post-Treatment of Erythrocytes with ALA, ALA Enantiomers, or MB and Treatment with DDS-NOH

The reduction of methemoglobin formation of ALA and its enantiomers was evaluated as follows: Erythrocytes were exposed to DDS-NOH (2.5 μg/mL) for 60 min at 37 °C as described by McMillan et al. [72]. These cells were then post-treated with 100 μM of ALA or R/S-ALA (0.1, 1, 5 and 10μM) for 60 min at 37 °C, as per Pandey et al. [73]. In addition, to assess the effect of MB, the erythrocytes suspension was exposed to DDS-NOH and after they were post-incubated for 30 min with MB (40 nM), as per Reilly et al. [74]. Cellular viability was analyzed prior to and after incubation.

### 4.7. Determination of Methemoglobin Content

Methemoglobin was determined according to Hegesh [75]. Methemoglobin content was evaluated in the buffered hemolysate through potassium cyanide-mediated conversion to cyanmethemoglobin, which absorbs at a wavelength of 632 nm. A dilution of the hemolysate, in which potassium ferricyanide (K_3_Fe (CN)_6_) was used to convert all possible forms of hemoglobin (Hb) to methemoglobin, was used as a reference solution. The methemoglobin content was measured in duplicate, and values less than 2% were considered normal.

### 4.8. Cell Culture and Sample Preparations to Comet Assay

Human venous blood was removed from several healthy volunteers who were abstainers from alcohol and tobacco (both sexes and ages of 20–45 years). Blood was obtained by venipuncture in heparin (5000 IU/mL). Briefly, 0.3 mL of venous blood was added to 4 mL of RPMI-1640 medium containing 20% of bovine calf serum and 50 μg/mL phytohemagglutinin. The mixture was then incubated at 37 °C in a 5% CO_2_ incubator for 24 h [76]. Lymphocytes were harvested with different drugs, as described below, and each sample was tested for viability using the trypan blue dye exclusion technique [76]. Only cell samples whose viability was over 90%, were measured by comet assay.

### 4.9. DNA Damage Using the Comet Assay

Lymphocytes were treated with DDS-NOH (7.5 μg/mL) and/or ALA (100 μM) and R/S-ALA (0.1, 1 and 10μM) for 3 hrs. Cell viability determination (90%) comet assay was then performed as described by Anderson [77]. To perform the comet assay, each sample was mixed with low melting-point agarose at 37 °C, to a final concentration of 0.5%. The mixture (100 μL) was pipetted onto slides pretreated with 1.5% normal-melting-point agarose, to retain the agarose cell suspension. The drop containing the cells was covered with a glass coverslip (24 mm × 24 mm) and left at 4 °C for 5 min. The coverslips were gently removed, and the slides were then ready for processing. The slides were treated with a lysis solution (2.5 M NaCl, 100 mM EDTA, 100 mM TRIS, 1% Triton X-100 and 10% DMSO, pH ~ 10.2) for 24 h at 4 °C. After protein removal, the slides were placed horizontally on an electrophoresis tray and the resultant nucleoids were immersed in electrophoresis buffer (300 mM NaOH and 100 mM EDTA, pH > 13) for 20 min at 4 °C to cleave the alkali-labile sites. The electrophoresis was then started using an electric field of 23 V/cm for 20 min. At the end of the process, the slides were gently removed from the tray and washed with distilled water for 5 min for neutralization. The slides were dehydrated by immersion for 3 min in absolute ethanol and were then air dried. Finally, the slides were stained with ethidium bromide (20 μg/mL) and viewed using fluorescence microscopy ZEISS AxioCam HRc with a green barrier filter 510–560 nm and 400 x coupled to a video camera. The cell images were analyzed using Tritek Cometscore Freeware 1.6 software. Registered parameters included the percent of DNA in the tail (Tail DNA %), tail length (TL), tail moment (TM) and Olive moment (OM) as markers of DNA damage. One hundred comets were scored randomly for each concentration employed. All experiments were performed in duplicate and hydrogen peroxide (H_2_O_2_, 200 μM) was employed as a positive control, which caused pronounced DNA damage and confirmed the accessibility of the cells to the tested chemicals. All steps described previously were carried out in a darkroom to prevent the interference of additional DNA damage.

### 4.10. Measurement of Intracellular Reactive Oxygen Species (ROS)

ROS production induced by DDS-NOH (2.5 and 7.5 μg/mL) in human erythrocytes was evaluated using 2′, 7′-Dichlorodihydrofluorescein diacetate (DCFH-DA). Erythrocyte suspensions were pretreated with ALA (100 and 1000 μM) for 1 h at 37 °C and subsequently, these cells were exposed to DDS-NOH or t-BHP for 30 min [78]. The t-BHP (200 μM), an organic peroxide widely used in a variety of oxidation processes, was used as a positive control [60]. Twenty minutes before the end of the exposure with DDS-NHOH, 10 μM DCFH-DA was added to the suspension and incubated for 30 min at 37 °C. Immediately, the DCF fluorescence intensity was measured by flow cytometry (FACSCanto, Becton Dickinson LSR II flow cytometer, San Jose, CA, USA) at an excitation wavelength of 488 nm, with a 530 nm emission filter [79,80]. When applied to intact cells, the nonionic, nonpolar, non-fluorescent DCFH-DA crosses cell membranes and is hydrolyzed enzymatically by intracellular esterases to form the intermediate to non-fluorescent 2,7-dichlorodihydrofluorescein (DCFH) that reacts with various ROS (including H_2_O_2_, OH•, and O_2_•−) and also by RNS (•NO and ONOO- to form 2′,7′-dichlorofluorescein (DCF), a highly fluorescent product [75]. Thus, some authors considered the DCFH as an effectiveprobe that not only measures the H_2_O_2_ in presence of cellular peroxidases but also determines the ONOO and HO• [81,82].

### 4.11. Reduced Glutathione Activity

Reduced glutathione was determined according to the methodology adapted from Ellman [83]. The reaction is based on the ability of this antioxidant to reduce 5,5-dithiobis-2-nitrobenzoic acid (DTNB) (Sigma-Aldrich) to 5-thio-2-nitrobenzoic acid (TNB). GSH was determined by spectrophotometry at 412 nm [84].

### 4.12. Superoxide Dismutase (SOD) Activity

Determination of SOD activity was performed according to the procedure recommended by McCord and Fridowich [85]. This method evaluated the ability of SOD to catalyze the conversion of O_2_- to H_2_O_2_ and O_2_. SOD activity was measured using UV spectrophotometry at a wavelength of 550 nm and was expressed in nmol/mL. t-BHP (200 μM) also was used as a positive control.

### 4.13. Trolox Equivalent Antioxidant Capacity

The Trolox equivalent antioxidant capacity (TEAC) was determined according to the methodology adapted and modified by Re [61]. In this assay, 2,2′-Azino-bis (3-ethylbenzothiazoline-6-sulfonic acid) diammonium salt (ABTS) (Sigma Aldrich) was incubated with potassium persulfate (Sigma Aldrich) to produce ABTS•+, a green/blue chromophore. Inhibition of ABTS•+ formation by antioxidants in the samples was expressed as Trolox equivalents. Thus, the total antioxidant activity of the sample was determined at 734 nm.

### 4.14. Determination of Lipid Peroxidation (MDA)

The dosage of thiobarbituric acid reactive substances (TBARS) is a method used to assess lipid peroxidation. The reaction of malondialdehyde (MDA) and other substances with thiobarbituric acid (TBA; Sigma-Aldrich), at pH 2.5 and 94 °C, forms the pink-colored MDA-TBA complex. The method was used as an indicator of oxidative stress and lipid peroxidation was determined at 535 nm.

### 4.15. Data Analysis

Data are reported as the mean ± S.E.M values. Data that were not normally distributed were analyzed non-parametrically. For the other parameters, statistically significant differences between groups were determined using Analysis of Variance (ANOVA) followed by the Tukey multiple comparison test. In all cases, the significance level adopted was 5% (α = 0.05).

### 4.16. Molecular Modeling

#### 4.16.1. Preparation of the Ligands and the Receptor

The 3D structures of the ligand molecules were drawn with the Avogadro [86] program, observing the configurations of the R-ALA and S-ALA isomers. The ligands were converted into more energetically-stable structures through quantum energy minimization calculations using the Gaussian 03 [87] program, with the DFT method and levels of the hybrid theory B3LYP [88] and 6-31G (d, p) [89] basis set. From these calculations, the energy values of the R and S configurations and the HOMO and LUMO molecular orbitals were obtained.

The 3D structure of hemoglobin used in the computational experiments was retrieved from the Protein data bank [90], with the code 4n8t6 [91]. Removal of water molecules and crystallization artifacts were performed using the Pymol [92] program. Polar Hydrogens and Gasteiger charges were added by AutoDock Tools (ADT) v1.5.6 [93].

#### 4.16.2. Molecular Docking

The molecular docking study was carried out with the Autodock 4.2.6/Vina [94] program, with the objective of evaluating the intermolecular interactions between the ligands and the hemoglobin molecule. The receptor molecule remained rigid and the ligands were allowed to rotate freely. All other parameters were kept as software defaults. The protein and ligand .pdb files were transformed into. pdbqt.

A blind docking approach was applied to fit the ligands to the receptor. The entire surface of the receptor molecule was covered to allow the ligands to freely bind to the protein’s access points, such that the central XYZ dimensions were constrained to 0.165 Å on the X axis, −0.709 Å on the Y axis and −0.533 Å on the Z axis, and dimensions in the XYZ size of 1261 Å, 126 Å and 126 Å, respectively.

Each ligand molecule was coupled 104 times to the receptor using an exhaustiveness value of eight. The ligand-receptor complex chosen was the one with the lowest energy, in Kcal.mol^−1^, which shows the highest chemical affinity (the more negative the score the better the affinity). The final visualization of the anchored structure was performed using Discovery Studio Visualizer 2.5.

## 5. Conclusions

The pretreatment with ALA, as a racemic mixture. was able to reduce methemoglobin levels, increase glutathione in erythrocytes and protect against DNA damage in lymphocytes. In regard to the enantiomers(R/S-ALA), they prevented at least 50% of MetHb formation and prevented DNA damage induced by DDS-NOH, but only the R-ALA enantiomer was able to partially reverse the formation of MetHb induced by DDS-NOH metabolites. Therefore, the racemic mixture of ALA was more effective in protecting against methemoglobin formation and DNA damage, while the R-ALA isomer was effective in protecting or partially reversing methemoglobin formation; data that were better explained by molecular docking, which showed that the binding energies of the R-ALA and S-ALA isomers differ, and due to a difference in curvature between the two isomers, the R-ALA isomer interacted with the HEME group, while the S-ALA interacted with the pro-44 amino acid residue. Thus, we conclude that the R-ALA isomer has a more stable format than the S-ALA isomer in binding to hemoglobin, which explains its better inhibitory action on methemoglobin formation in vitro. These findings show that ALA could be used in future therapies in patients who use dapsone chronically, including leprosy patients. Moreover, ALA may also be of benefit in patients suffering from high dapsone-induced methemoglobinemia who cannot tolerate methylene blue due to glucose-6-phosphate deficiency [95]. 

## Figures and Tables

**Figure 1 ijms-24-00057-f001:**
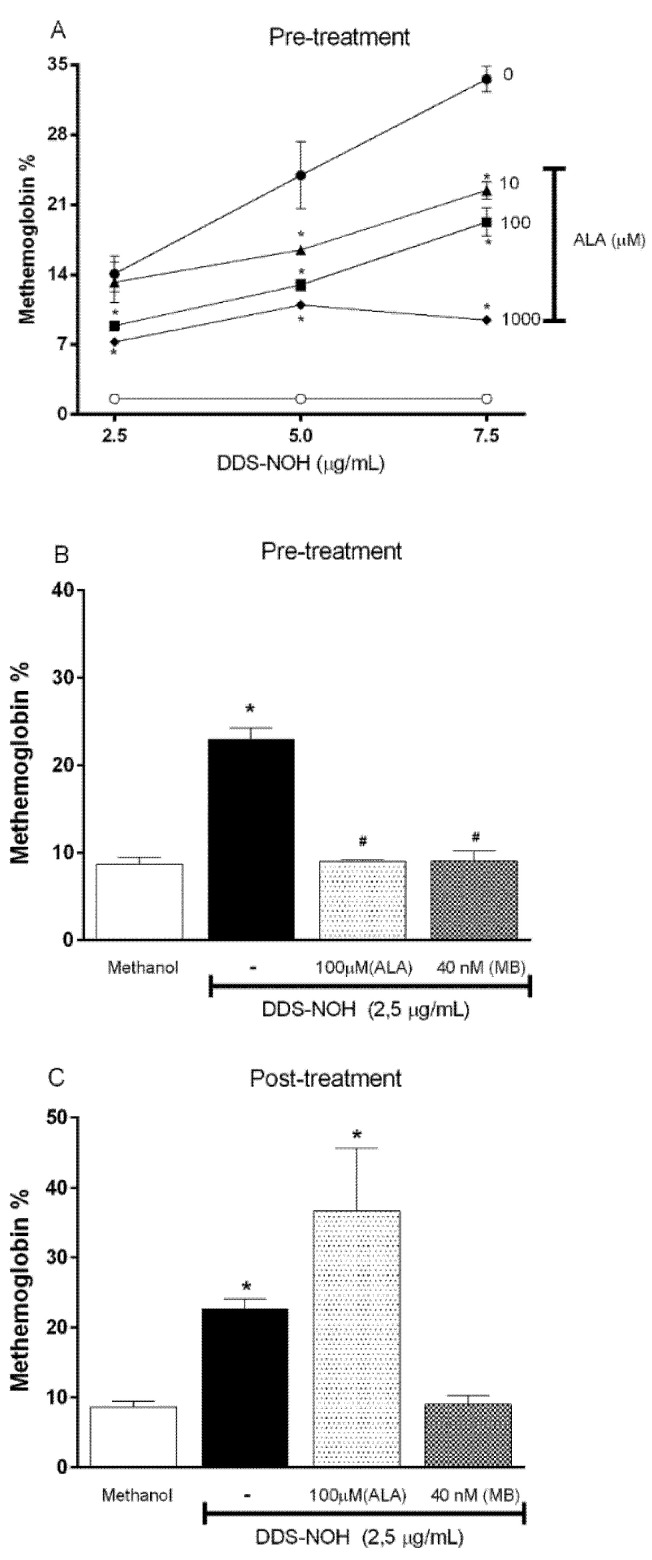
Effect of pre- or post-treatment with racemic Alpha-Lipoic Acid (ALA) on methemoglobin formation induced by DDS-NOH and its comparison to MB. (**A**) Pretreatment with different concentrations of ALA; (**B**) Comparison between pretreatment of ALA and MB and (**C**) Comparison between post-treatment of ALA and MB on methemoglobin formation induced by DDS-NHOH. For this, erythrocytes were pretreated with different concentrations of ALA (10, 100 and 1000 µM), or post-treated (100 µM), for 1 h at 37 °C, then these cells were incubated with DDS-NOH (2.5 µg/mL) for 1 h at 37 °C. Data are reported as means ± S.E.M from three independent experiments done in triplicate. * *p* ≤ 0.05 compared to methanol; # *p* < 0.05 compared to the DDS-NOH group.

**Figure 2 ijms-24-00057-f002:**
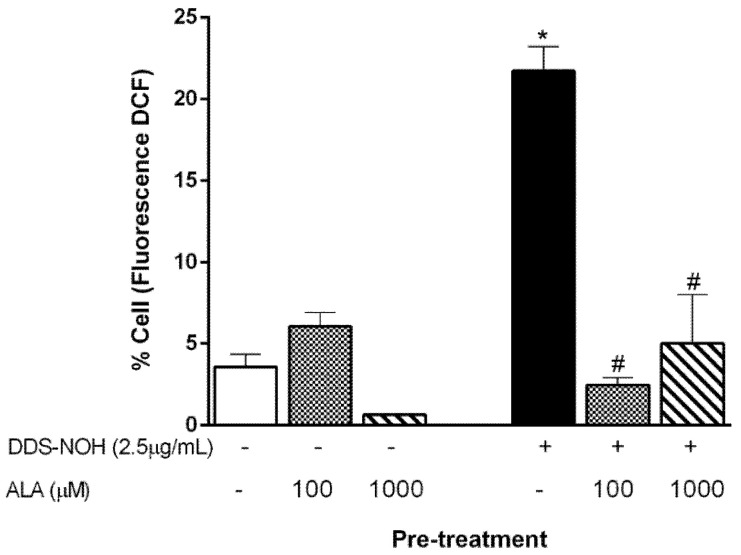
Effect of pretreatment with ALA on reactive oxygen species (ROS) generation induced by DDS-NOH. Erythrocytes were pretreated with ALA (100 and 1000 µM) for 1 h at 37 °C and incubated for 30 min with DDS-NOH (2.5 µg/mL). ROS production was measured as dichlorofluorescein (DCF) fluorescence. Values are means ± S.E.M. * *p* < 0.05 compared to methanol. # *p* < 0.05 compared to the DDS-NOH group.

**Figure 3 ijms-24-00057-f003:**
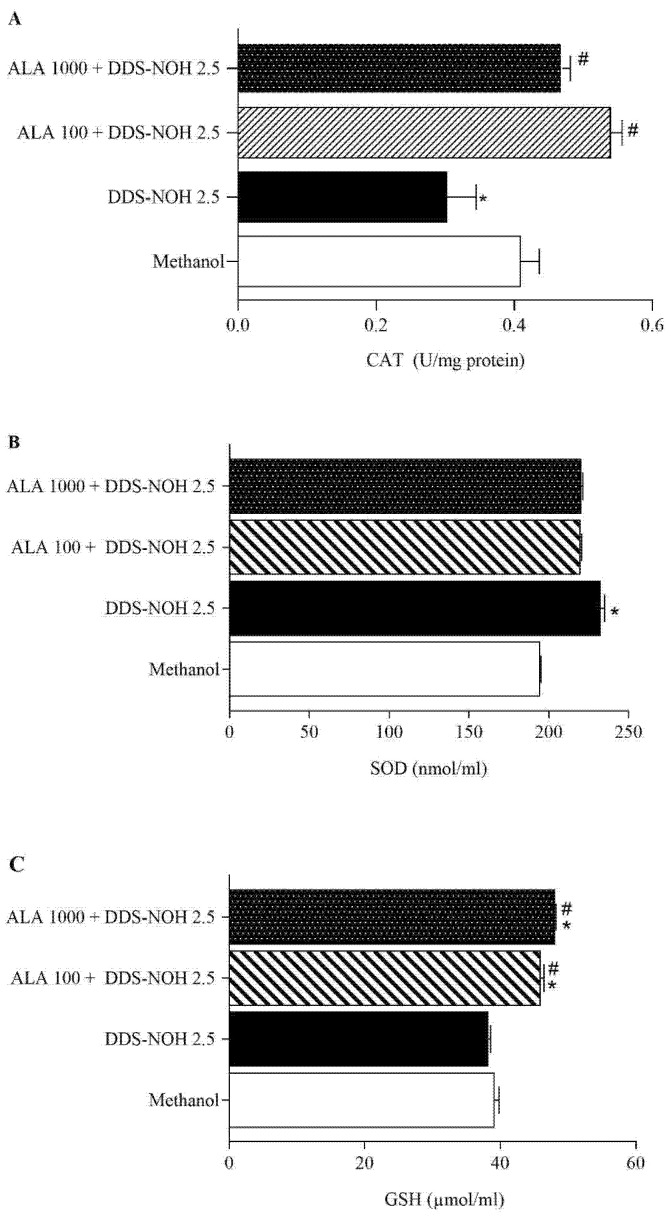
Effect of pretreatment with ALA on CAT and SOD activities and GSH production. (**A**) Catalase activity with different concentrations of ALA (**B**) SOD activity with different concentrations of ALA (**C**) Glutathione activity with different concentrations of ALA Erythrocytes were pretreated with ALA (100 and 1000 µM) for 1 h at 37 °C and incubated for 30 min with DDS-NOH (2.5 µg/mL). Results are expressed as mean ± S.E.M. * *p* < 0.05 compared to methanol; # *p* < 0.05 compared to the DDS-NOH group.

**Figure 4 ijms-24-00057-f004:**
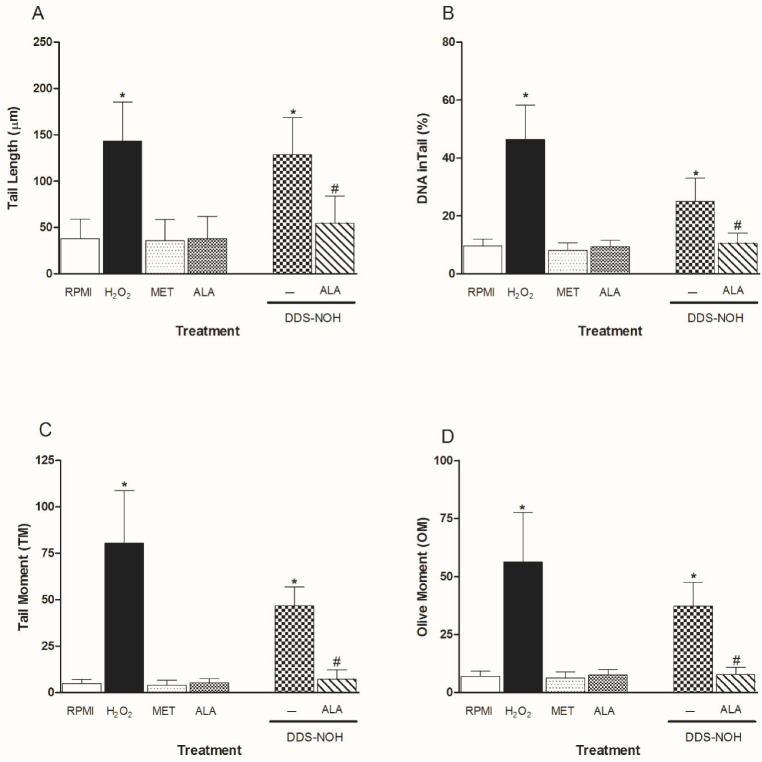
Effect of pretreatment with ALA on DNA damage induced by DDS-NOH. Tail length (µm—**A**), DNA in tail (%—**B**) tail moment (TM—**C**) and Olive moment (OM—**D**) were used as a marker of DNA damage in lymphocytes using comet assay. Values are means ± S.E.M. * *p* < 0.05 compared to methanol. # *p* < 0.05 compared to the DDS-NOH group.

**Figure 5 ijms-24-00057-f005:**
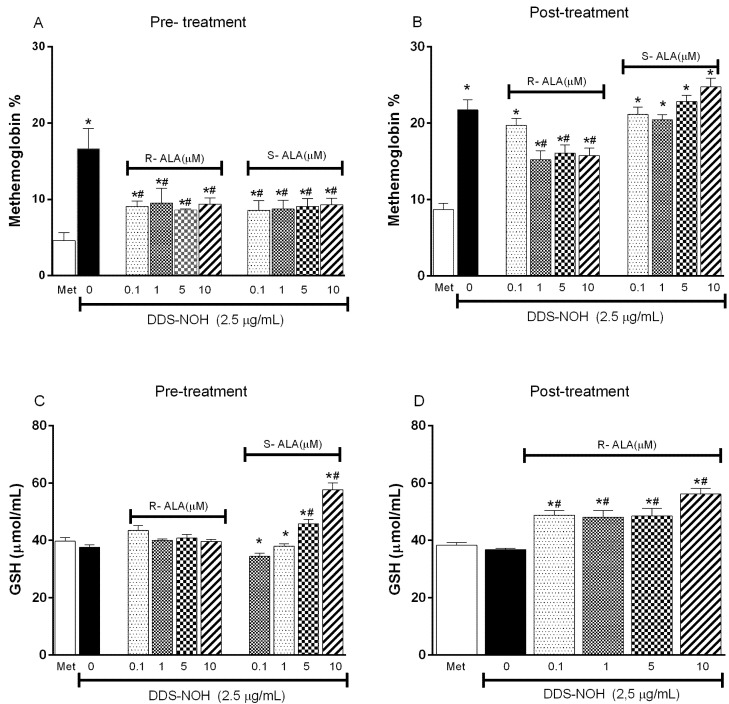
Effect of the pre- or post-treatment with R-ALA or S-ALA on methemoglobin formation and GSH production induced by DDS-NOH. (**A**) pre-treated ALA methemoglobin levels in different concentrations of R and S-ALA; (**B**) post-treated R and S-ALA methemoglobin levels in different contractions; (**C**) glutathione levels in samples pretreated with different R and S-ALA concentrations; (**D**) glutathione levels in samples post treated with different R and S-ALA concentrations. Erythrocytes were pretreated or post-treated with R or S-alpha Lipoic Acid (R-ALA or S-ALA, 0.1, 1; 5 and 10 µM) for 1 h at 37 °C and incubated for 30 min with DDS-NOH (2.5 µg/mL). Values are means ± S.E.M. * *p* < 0.05 compared to methanol. # *p* < 0.05 compared to the DDS-NOH group.

**Figure 6 ijms-24-00057-f006:**
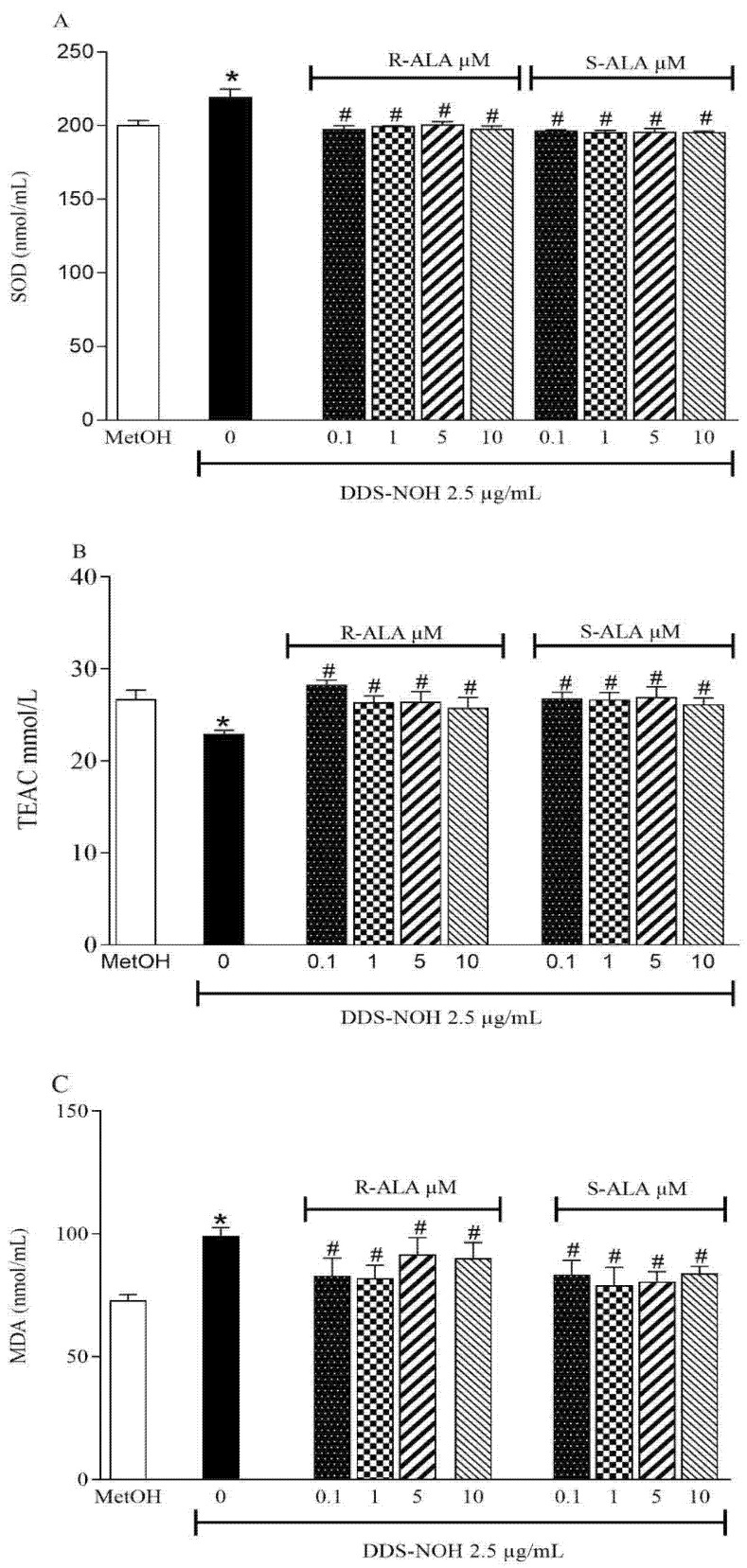
Effect of pretreatment with R-ALA or S-ALA on oxidative parameters induced by DDS-NOH. (**A**) pre-treated ALA SOD activity levels in different concentrations of R and S-ALA; (**B**) pretreated R and S-ALA TEAC levels in different contractions; (**C**) MDA levels in samples pretreated with different R and S-ALA concentrations. Erythrocytes were pretreated with R and S-alpha Lipoic Acid (R-ALA or S-ALA, 0.1, 1, 5 and 10 µM for 1 h at 37 °C and incubated for 30 min with DDS-NOH (2.5 µg/mL). Values are means ± S.E.M. * *p* < 0.05 compared to methanol. # *p* < 0.05 compared to the DDS-NOH group.

**Figure 7 ijms-24-00057-f007:**
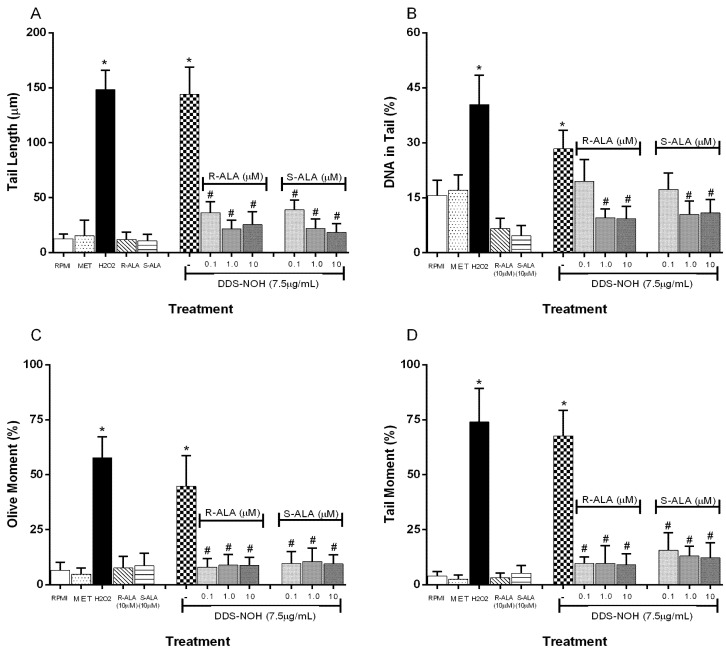
Effect of pretreatment with R-ALA or S-ALA on DNA damage induced by DDS-NHOH. Tail length (µm—**A**), DNA in tail (%—**B**) tail moment (TM—**C**) and Olive moment (OM—**D**) were used as a marker of DNA damage in lymphocytes using comet assay. All values are depicted as mean ± SE. Values are means ± S.E.M. * *p* < 0.05 compared to methanol. # *p* < 0.05 compared to the DDS-NOH group.

**Figure 8 ijms-24-00057-f008:**
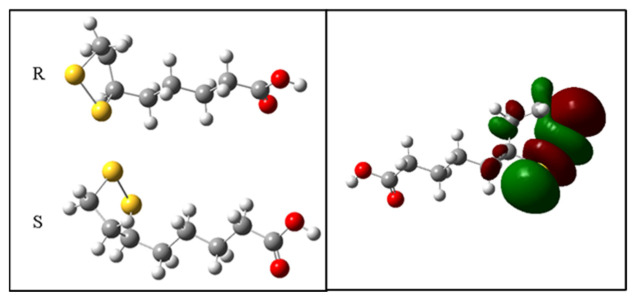
Electronic density of the boundary orbitals of the R-ALA and S-ALA isomers.

**Figure 9 ijms-24-00057-f009:**
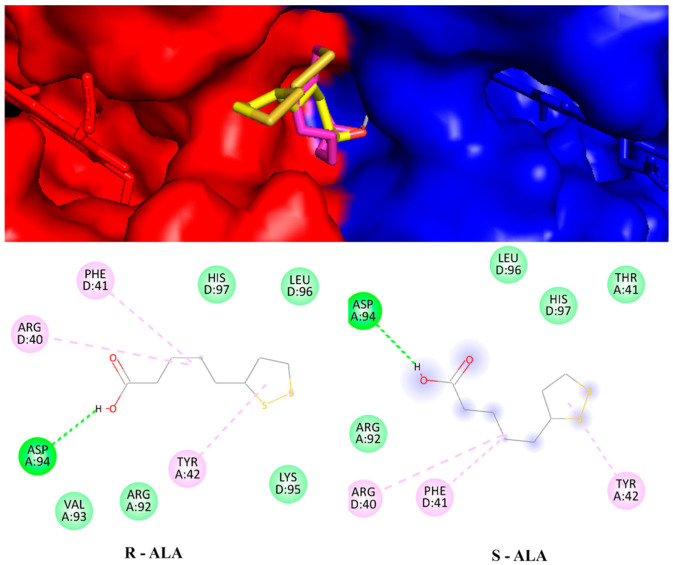
Hemoglobin structure is anchored with the ligands R-ALA and S-ALA between the A (blue) and D (yellow) chains and their respective interaction residues.

**Figure 10 ijms-24-00057-f010:**
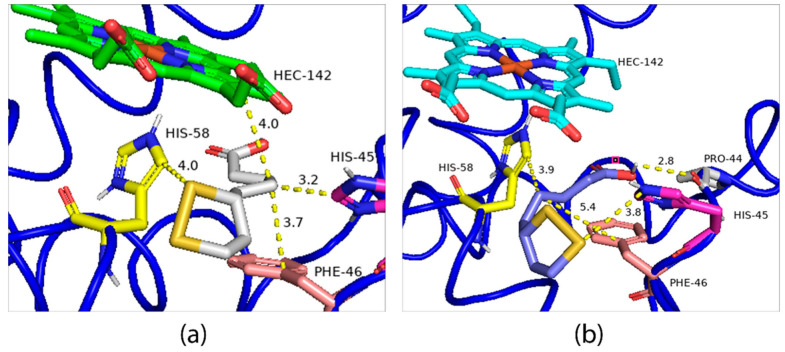
Lower energy R-ALA (**a**) and S-ALA (**b**) configurations obtained through molecular docking.

**Table 1 ijms-24-00057-t001:** HOMO, LUMO and GAP values obtained through quantum calculations.

Comp	HOMO (eV)	LUMO (eV)	GAP (eV)
1	−5.58	−0.97	4.61
2	−5.56	−1.00	4.55

## Data Availability

The datasets generated for this study are available on request to the corresponding author.

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
