# Peer review of "Alpha-Lipoic Acid and Its Enantiomers Prevent Methemoglobin Formation and DNA Damage Induced by Dapsone Hydroxylamine: Molecular Mechanism and Antioxidant Action"

_ijms, 2022, doi:10.3390/ijms24010057_

Round 1

Reviewer 1 Report

This Ms by Kaio Murilo Monteiro Espíndola et al., is informative ans sounds good in general. The introduction and discussion sections are well written and provide well-focused information. In general, the Ms is well written and the conclusion are consistent with the data provided.

However, some minor points have to be addressed as indicated below.

1) Page 4: The title and legend for “figure 1” is missing.

2) Page 6: Figure 3A is missing (only 3B and 3C are presented). Moreover, the tags 3A, 3B and 3C have to be indicated on the figure.

3) Page 10 line 169: (potential mistake) it seems that 101µM has to be change in 10µM

Author Response

Dear Reviewer

Thanks for your comments and suggestions. We have included the title and legend for figure 1, changed figure 3 (3A, 3B and 3C) and corrected the text as requested.

Reviewer 2 Report

The manuscript presents interesting data but the presentation is far from perfect. Data concerning ROS level in erythrocytes are not presented (they should be shown and discussed, otherwise part 4.10 should be removed). One Figure is not complete, some Figures and legends require amendments/completion. Linguistic amendment is necessary.

Detailed remarks:

The abbreviations system used in Abstract are not explained. DDS-NHOH, MET, TEAC and T-BARS are  not explained.

Line 29, please use “.” as a decimal separator

Lines 36 and 155: “increased of SOD, decreased of TEAC”, please change to “increase of SOD, decrease of TEAC”

Line 47: “is used to in non-infectious inflammatory diseases”, “used, too”?

Line 109: “ALA efficiency inhibited’, perhaps “ALA efficiently inhibited”

Line 112: “even at a concentration of MET was at least 2000-fold”, please delete “was”

Line 116: “T-BHP”, please explain

Lines 116/117: “erythrocytes induced”, please re-formulate

Fig. 127: the term “reverted” is not appropriate in the case of a pretreatment

Figure 3: Panels are not labeled (A,B,C). Panel A is missing. What is the control level? “Methanol”?

Figures 5 and 6: What is the meaning of “Met”  and “0” on the axis of abscissae? It should be explained.

Figure 6: Please correct “0,1” to “0.1”. TEAC should be expressed in mmol/L, not mM/L  (mM = mmol/L). MDA should be expressed in nmol/mL, not nm/mL

Lines 157/158: Why higher concentrations of  R-ALA were not effective?

Figures: It should be indicated what was the reference for statistic comparisons.

Line 169: 101 microM?

Line 266: “Methemoglobin uptake” is not an appropriate term

Line 274: “could protect the formation of methemoglobin”, perhaps “protect against the formation…”

Line 281: “such as lysine residues”, perhaps “via lysine residues”

Line 297: “concentrations of 50 to 600 mg”, these are doses, not concentrations

Lines 330/331: what is “glutaraldeutinase”?

Lines 375/376: tert-butyl hydroperoxide, not tetrabutylhydroperoxide

Line 550: “concentration” rather than “activity”

Author Response

Dear Reviewer

Thanks for your comments and suggestions. As requested, we have included the abbreviations in the abstract and elsewhere in the article, corrected the text and figures (the decimal separator and corrected the units), included the statistical parameters in the legends, and figure 3 has been reformulated. Dapsone is a drug used in infectious and non-infectious diseases, and we used methanol as a negative control and methylene blue (MB) as a positive control in the analysis of methemoglobin.

Thanks again

Round 2

Reviewer 2 Report

The amended manuscript is acceptable.